# Risk for Irritable Bowel Syndrome in Patients with Helicobacter Pylori Infection: A Nationwide Population-Based Study Cohort Study in Taiwan

**DOI:** 10.3390/ijerph17103737

**Published:** 2020-05-25

**Authors:** Chia-Ming Liang, Chih-Hsiung Hsu, Chi-Hsiang Chung, Chao-Yang Chen, Lin-Yin Wang, Sheng-Der Hsu, Pi-Kai Chang, Zhi-Jie Hong, Wu-Chien Chien, Je-Ming Hu

**Affiliations:** 1Division of General Surgery, Tri-Service General Hospital, National Defense Medical Center, Taipei 11490, Taiwan; kevin.magic77@gmail.com (C.-M.L.); f1233j@yahoo.com.tw (S.-D.H.); lgf670822@mail.ndmctsgh.edu.tw (Z.-J.H.); 2Graduate Institute of Medical Sciences, National Defense Medical Center, Taipei 11490, Taiwan; chihhsung@gmail.com (C.-H.H.); pencil8850@hotmail.com (P.-K.C.); 3Teaching Office, Tri-Service General Hospital, National Defense Medical Center, Taipei 11490, Taiwan; 4School of Public Health, National Defense Medical Center, Taipei 11490, Taiwan; g694810042@gmail.com; 5Department of Medical Research, Tri-Service General Hospital, National Defense Medical Center, Taipei 11490, Taiwan; 6Taiwanese Injury Prevention and Safety Promotion Association, Taipei 11490, Taiwan; 7Division of Colorectal Surgery, Department of surgery, Tri-service General Hospital, National Defense Medical Center, Taipei 11490, Taiwan; cartilage77@yahoo.com.tw; 8Department of Surgery, Taoyuan Armed Forces General Hospital, Taoyuan 32551, Taiwan; kate79971990@gmail.com; 9Graduate Institute of Life Sciences, National Defense Medical Center, Taipei 11490, Taiwan; 10School of Medicine, National Defense Medical Center, Taipei 11490, Taiwan

**Keywords:** *Helicobacter pylori* infection (*H. pylori* infection), National Health Insurance Research Database (NHIRD), irritable bowel syndrome (IBS), retrospective cohort study

## Abstract

Background: The association between *Helicobacter pylori* (*H. pylori*) infection and the risk of developing irritable bowel syndrome (IBS) has yet to be investigated; thus, we conducted this nationwide cohort study to examine the association in patients from Taiwan. Methods: A total of approximately 2669 individuals with newly diagnosed *H. pylori* infection and 10,676 age- and sex-matched patients without a diagnosis of *H. pylori* infection from 2000 to 2013 were identified from Taiwan’s National Health Insurance Research Database. The Kaplan–Meier method was used to determine the cumulative incidence of *H. pylori* infection in each cohort. Whether the patient underwent *H. pylori* eradication therapy was also determined. Results: The cumulative incidence of IBS was higher in the *H. pylori*-infected cohort than in the comparison cohort (log-rank test, *p* < 0.001). After adjustment for potential confounders, *H. pylori* infection was associated with a significantly increased risk of IBS (adjusted hazard ratio (aHR) 3.108, *p* < 0.001). In addition, the *H. pylori*-infected cohort who did not receive eradication therapy had a higher risk of IBS than the non-*H. pylori*-infected cohort (adjusted HR 4.16, *p* < 0.001). The *H. pylori*-infected cohort who received eradication therapy had a lower risk of IBS than the comparison cohort (adjusted HR 0.464, *p* = 0.037). Conclusions: Based on a retrospective follow-up, nationwide study in Taiwan, *H. pylori* infection was associated with an increased risk of IBS; however, aggressive *H. pylori* infection eradication therapy can also reduce the risk of IBS. Further underlying biological mechanistic research is needed.

## 1. Introduction

Irritable bowel syndrome (IBS), a functional disorder of the gastrointestinal tract, is formalized in the Rome criteria, which include chronic abdominal pain and altered bowel habits [1]. IBS is the most frequently diagnosed GI disorder [2] and the prevalence of IBS worldwide is 10% to 15% of the population [3]. In Taiwan, the prevalence of IBS (2003) was 22.1% according to the Rome II criteria, and no sex difference was found between subjects with and without IBS [4].

The pathophysiology of IBS is multifactorial and remains uncertain [5]. Recently, several studies have proposed new hypotheses of IBS pathophysiology, including genetics, psychosocial dysfunction (brain-gut axis dysfunction) and inflammation [6,7,8,9]. Based on the inflammation hypothesis, epithelial cells of the inflamed intestinal mucosa are more permeable and lead to fluid leakage into the intestine. Compared with non-IBS subjects, IBS subjects have increased mast cells, lymphocytes, TNF alpha, IL-6, LIF, NGF, and IL-1 beta in the intestinal mucosa [9]. Therefore, *Helicobacter pylori* (*H. pylori*) infection has a potential pathogenic role in IBS given the significant overlaps between IBS and dyspepsia [10]. Approximately more than half of the world’s population is estimated to be infected with *H. pylori,* which is a widespread gram-negative bacterium that colonizes the human gut [11]. *H. pylori* infection irritates immune neutrophils and lymphocytes in gut epithelial cells and liberates cytokines, including IL-1, IL-6, IL-8, IL-12, TNF-α and IFN-γ [12,13]. Some studies have highlighted an increase in *H. pylori* infection rates in patients with IBS compared to healthy groups [14,15]. However, other studies have disputed this finding and found no association between *H. pylori* infection and IBS [16,17]. The inconsistent results were considered to be due to IBS caused by the interaction of multiple factors. Despite multiple investigations, data are conflicting, and no abnormality has been found to be specific for this disorder. Current evidence does not support an association between *H. pylori* infection and IBS [16]. Therefore, we used the randomized longitudinal health insurance dataset (LHID) selected from the National Health Insurance Research Database (NHIRD) to find IBS in the *H. pylori*-infected population in Taiwan.

## 2. Materials and Methods

### 2.1. Data Source

In 1995, the government of Taiwan implemented a single-payer universal health insurance system, the Taiwan National Health Insurance (NHI) program, which covers more than 99% of the 23 million residents in Taiwan and has been widely used in academic studies [18]. The NHIRD contains comprehensive, high-quality information, including epidemiological research, and information on diagnoses, prescription use, and health-care information, including inpatient/ambulatory claims, prescription claims, demographic data and hospitalization [19,20,21]. Data for our cohort study were obtained from the NHIRD, which has multiple data sources, is a powerful research engine with enriched dimensions and serves as a guiding light for real-world evidence-based medicine in Taiwan [22]. The International Classification of Disease, Ninth Revision, Clinical Modification (ICD-9-CM) was used to define the diagnostic codes.

### 2.2. Sample Participants

As shown in Figure 1, we identified patients aged 14 years or older with a newly diagnosed *H. pylori* infection (ICD-9-CM codes 041.86) in the period from 2000 to 2013. The ICD-9-CM diagnosis of *H. pylori* infection was based on endoscopic biopsy urease testing or non-invasive tests, such as urea breath testing. The diagnosis of *H. pylori* infection was confirmed at least twice. *H. pylori* infection is usually treated with triple eradication therapy (amoxicillin/metronidazole, levofloxacin/clarithromycin and a proton pump inhibitor) for at least two weeks to help prevent the bacteria from developing resistance to one particular antibiotic. Those with a documented *H. pylori* infection before 1 January 2000 or with incomplete medical information were excluded to ensure the data were from individuals with a first diagnosis of *H. pylori* infection. In Taiwan, the diagnoses of IBS (ICD-9-CM code: 564. 1) were made by board-certified gastroenterologists, and were diagnosed according to the serial versions of the Rome criteria [23]. The NHI administration randomly reviews the records of 1 in 100 ambulatory care visits and 1 in 15 inpatient claims to verify the accuracy of the diagnosis [24]. We also excluded people with a history of IBS. The comparison cohort consisted of patients randomly selected from the patient database without a history of *H. pylori* infection. For each identified patient with *H. pylori* infection, four comparison patients were randomly identified and frequency-matched according to the age, sex, and year of index date for the non-*H. pylori*-infected cohort.

### 2.3. Variables of Interest

The sociodemographic variables used in this study comprised age, sex and urbanization level. The level of urbanization was divided into four levels (Level 1 is the most urbanized and level 4 is the least urbanized) based on the NHRID report. We defined baseline comorbidities, coronary artery disease (ICD-9-CM codes 410-414), cerebrovascular accident (ICD-9-CM codes 430-438), hypertension (ICD-9-CM codes 401-405), hyperlipidemia (ICD-9-CM codes 272), diabetes (ICD-9-CM codes 250), end-stage renal disease (ICD-9-CM codes 585.6), asthma (ICD-9-CM codes 493) and medication use of tetracycline (drug codes: AC04963100, AC049631G0, AC12059100, AC12639100, AC15791100, AC157911G0, AC22572100, AC225721G0), minocycline (drug codes: A032761100, A033214100, A035969100, A036813100, A043111100, AB40644100, AC33471100, AC35868100, AC36266100, AC36281100, AC36667100, AC36815100, AC36940100, AC38761100, AC39074100, AC39600100) and doxycycline (drug codes: A009397100, AC07233100, AC12782100, AC16227100, AC19254100, AC192541G0, AC23648100, AC236481G0, AC24085100, AC31219100, AC34900100, AC35692100, AC356921G0).

### 2.4. Statistical Analysis

The distribution of sociodemographic data and comorbidities were compared between the *Helicobacter pylori* infection (HPI) cohort and the comparison cohort using the chi-squared test to examine categorical variables and the t-test to examine continuous variables. Kaplan–Meier analysis was performed to estimate IBS in these two cohorts, and the log-rank test examined the difference between the curves. We computed the incidence density rate of IBS (per 100,000 person-years) at follow-up for each cohort. Univariable and multivariable Cox proportion hazard regression models were used to examine the effect of HPI on the risk of IBS, shown as a hazard ratio (HR) with a 95% confidence interval (CI). The multivariable models were adjusted for age, sex, urbanization level and all comorbidities. Further analysis was performed to assess the effect of the dose response of *H. pylori* infection to the antibiotic medicine that was prescribed on the risk of IBS. All statistical analyses were performed using SPSS software (Version 22.0; SPSS Inc., Chicago, IL, USA). The comparisons used the significance level of 0.05 for two-sided testing.

### 2.5. Data Availability

Data are available from the National Health Insurance Research Database (NHIRD) published by the Taiwan National Health Insurance (NHI) Administration. Due to legal restrictions imposed by the government of Taiwan in relation to the “Personal Information Protection Act”, data cannot be made publicly available. Requests for data can be sent as a formal proposal to the NHIRD (http://www.mohw.gov.tw/cht/DOS/DM1.aspx?f_list_no=812).

### 2.6. Ethics Statement

This study was approved by the Institutional Review Board of the Tri-Service General Hospital (TSGH IRB No. 2-105-05-082; Taipei, Taiwan).

## 3. Results

Table 1 shows the demographic characteristics of both cohorts. The mean age was 57.71 ± 17.01 years in the study cohort and 57.70 ± 18.40 years in the control cohort. The difference was not significant (*p* = 0.593). In addition, there was no significant difference in sex or age for both groups. Regarding insurance premiums (in New Taiwan dollars (NT$)) in both cohorts, almost all of the enrolled patients were in the <18,000 group (98.25%), followed by the 18,000 to 34,999 group (1.44%) and the >35,000 group (0.31%). There was no significant difference in the insurance premiums (NT$) in either group (*p* = 0.136). In terms of the comorbidity, patients with *H. pylori* infection had lower rates of coronary artery disease (CAD), cerebrovascular accident (CVA), hypertension (HTN), hyperlipidemia, diabetes mellitus (DM) and asthma than the control patients. The CCI_R value was 1.27 ± 0.77 in the study cohort and 0.27 ± 0.69 in the control cohort (*p* < 0.001). In addition, more individuals in the study cohort than in the control cohort lived in southern and eastern Taiwan and lower urbanized areas, and received therapy in local hospitals (*p* < 0.001).

The IBS rate was calculated by the Kaplan–Meier method (Figure 2). The results showed that the study cohort had a significantly higher IBS rate than the control cohort (log-rank test (p < 0.001)). Table 2 shows the Cox regression analysis of the risk factors for IBS. After adjusting for *H. pylori* infection, sex, age group, insurance premium and pre-existing comorbidities, including sleep apnea, HTN, and only *H. pylori* infection (aHR = 3.108, 95% confidence interval (CI) = 1.934–4.995, *p* < 0.001), patients in the study cohort showed an increased risk of IBS diagnosis compared with the control cohort. In addition, patients with CVA had a decreased risk of developing IBS than those without CVA (aHR = 3.121, *p* < 0.001). Patients with other comorbidities, age groups and insurance premiums, as well as other chronic diseases, were not significantly associated with IBS diagnosis according to the hazard ratios (all *p* > 0.05).

In the subgroup analysis comparing patients with and without *H. pylori* infection (Table 3), the overall incidence of IBS was 109.5 per 100,000 person-years in the study cohort and 33.83 per 100,000 person-years in the control cohort. Female and male *H. pylori*-infected patients showed an increased risk of developing IBS (total aHR = 3.108, *p* < 0.001; 3.074 in the corresponding age group of females, *p* < 0.001; 3.228 in the corresponding age group of males, *p* = 0.003). In the age group analysis, patients with *H. pylori* infection in the 44 to 64-year age group were independently associated with an increased risk following IBS diagnosis compared with patients without *H. pylori* infection (aHR= 4.971, *p* < 0.001). In the insured premium group analysis, *H. pylori*-infected patients in the <18,000 group showed an increased risk of developing IBS (aHR = 3.108, *p* < 0.001). In the pre-existing condition group analysis, *H. pylori*-infected patients with CAD, CVA, HTN or DM displayed an increased risk of developing IBS (aHR = 3.268, *p* < 0.001 in CAD; aHR = 3.605, *p* < 0.001 in CVA; aHR= 4.702, *p* < 0.001 in HTN; aHR = 15.952, *p* < 0.001 in DM). In the season group analysis, patients infected with *H. pylori* in the summer and autumn groups showed an increased risk of developing IBS (aHR = 4.035, *p* = 0.001 in summer; aHR = 3.594, *p* = 0.009 in autumn). In the level of care group analysis, *H. pylori*-infected patients in the hospital center and regional hospital groups displayed an increased risk of developing IBS compared with those in clinics (aHR = 5.402, *p* = 0.003 in hospital center; aHR = 3.462, *p* < 0.001 in regional hospital). Patients with *H. pylori* infection were associated with a higher risk of IBS than patients without *H. pylori* infection, and patients with *H. pylori* infection who received aggressive eradication therapy had a decreased risk of IBS compared with patients with *H. pylori* infection who did not receive eradication therapy (adjusted HR 0.464; 95% CI 0.148–0.963, *p* = 0.037) (Table 4). 

## 4. Discussion

This is the first large-scale, population-based study with long-term follow-up on the association between *H. pylori* infection and the risk of IBS. After adjusting for covariates, the overall adjusted HR was 3.108 (95% CI: 1.934–4.995, P < 0.001). In other words, adult patients with *H. pylori* infection had a nearly 3.1-fold increased risk of developing IBS than those without *H. pylori* infection. The Kaplan–Meier analysis revealed that the study subjects had a significantly higher 14-year IBS rate than the controls after adjusting for potential confounders; this is further strong evidence for a potential association between IBS and *H. pylori* infection.

*H. pylori* can produce bacterial urease, which hydrolyses in gastric luminal urea to form ammonia, which helps neutralize gastric acid and form a protective cloud around the organism, enabling it to penetrate the gastric mucus layer [25]. Several studies have reported an association between *H. pylori* infection and the incidence of gastric cancer and colorectal cancer [26,27]. Diagnostic testing for *H. pylori* includes endoscopic biopsy urease testing, urea breath testing, stool antigen assay, and serological ELISA testing [28]. *H. pylori* infection is usually treated with at least two different antibiotics at once to help prevent the bacteria from developing resistance to one particular antibiotic [29]. Triple eradication therapy and bismuth quadruple treatment regimens are popular in Taiwan.

The pathogenesis of IBS is still uncertain. In our study, we tried to find a relationship between IBS and *H. pylori* infection. It seems that chronic inflammation caused by *H. pylori* infection is related to the pathogenesis of atrophic gastritis, intestinal metaplasia and peptic ulcers [30]. *H. pylori* infection may increase inflammatory marker levels [31], increase mast cell activation, and then affect the gastric mucosa and nerve remodeling [32,33,34], causing visceral hypersensitivity symptoms such as those of IBS [35,36,37,38].

According to our study, patients with *H. pylori* infection who did not receive eradication therapy had a higher risk of IBS than the non-*H. pylori*-infected cohort. (adjusted HR 4.16; 95% CI 2.508–6.900). Patients with *H. pylori* infection who received eradication therapy had a lower risk of IBS than the untreated group (adjusted HR 0.464; 95% CI 0.148–0.963). In the long term observation, the risk of IBS is reduced in the H. pylori infection-treated population. There seems to be some connections between *H. pylori* infection and IBS. Clinically, it is also possible to add a supplemental *H. pylori* test for drug treatment to reduce the symptoms of IBS.

The association between IBS and *H. pylori* has been challenged. For example, *H. pylori* is thought to commonly influence the upper GI tract rather than the lower GI tract. Additionally, findings by Xiong indicate that there is no relationship between *H. pylori* infection and IBS [39]. However, although this multi-center retrospective study considered eradication medicine resistance, the cohort was only followed up for one year, and there was no follow-up after second-line drugs were administered. Other data have shown that the association could be fortuitous given the widespread prevalence of *H. pylori* infection globally [40]. Clinically, *H. pylori* infection drugs fail or resistance develops in approximately 10% of patients [41], and there are other many reasons for failure. Perhaps our big data research can reduce these factors.

This study has a number of strengths. First, the sample size was very large, which enhanced the statistical power of the data. We used stratified analyses based on the confounding factors of age, comorbidities, and a wide range of demographic characteristics. Second, because we used a nationwide database with a very high coverage rate, almost all patients’ follow-up data were available. Third, the population-based data were considered representative of the general population in Taiwan.

Our study also had some limitations. First, this was a retrospective cohort study; therefore, it has lower statistical quality. Bias from unknown confounders and errors of primary records may have affected our results, and a well-designed prospective, randomized, controlled study is needed to help establish causal relationships. Second, the NHIRD does not report information such as clinical symptoms, signs, or laboratory data, which may help distinguish different types of IBS for further analysis. 

## 5. Conclusions

Patients with *H. pylori* infection have a higher proportion of IBS, and aggressive *H. pylori* infection eradication therapy can reduce the risk of IBS. Further underlying biological mechanistic research is needed. Clinically, IBS patients seem to be able to undergo routine *H. pylori* diagnostic tests to determine the direction of treatment.

## Figures and Tables

**Figure 1 ijerph-17-03737-f001:**
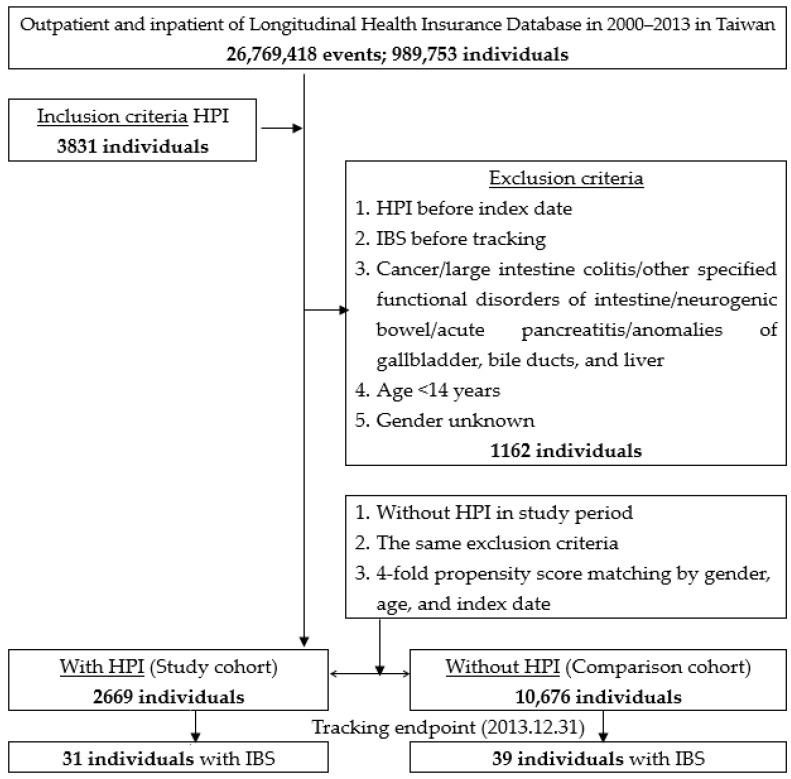
The flowchart of the study sample selection from the National Health Insurance Research Database in Taiwan. HPI: *Helicobacter pylori* infection.

**Figure 2 ijerph-17-03737-f002:**
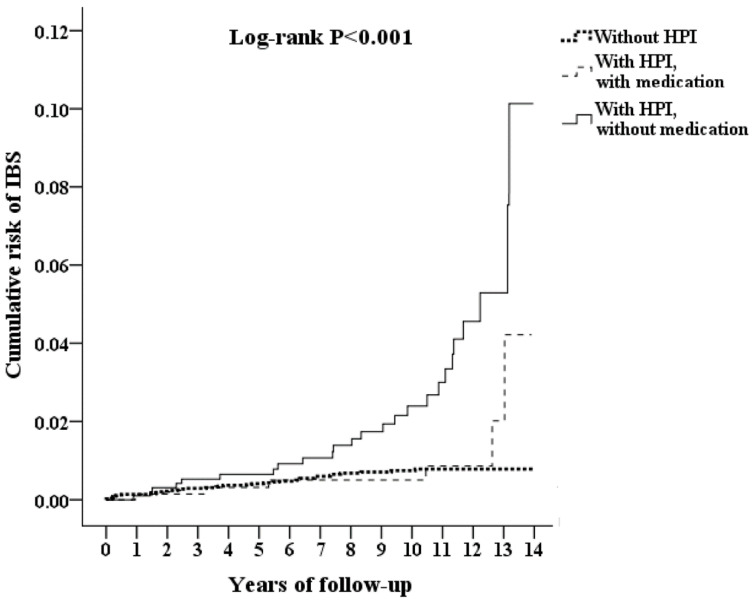
Kaplan–Meier for cumulative risk of IBS in patients aged 14 and over stratified by *Helicobacter pylori* infection (HPI) with the log-rank test. Without HPI vs. With HPI, without medication: Log-rank P < 0.001. Without HPI vs. With HPI, with medication: Log-rank P = 0.479. With HPI, without medication vs. With HPI, with medication: Log-rank P < 0.001.

**Table 1 ijerph-17-03737-t001:** Characteristics of the study population at the baseline.

HPI	Total		With		Without		*P*
Variables	n	%	n	%	n	%	
**Total**	13,345		2669	20.00	10,676	80.00	
**Gender**							0.999
Male	8790	65.87	1758	65.87	7032	65.87	
Female	4555	34.13	911	34.13	3644	34.13	
**Age (years)**	57.54 ± 18.13		57.71 ± 17.01		57.70 ± 18.40		0.593
**Age group (years)**							0.999
<44	3350	25.10	670	25.10	2680	25.10	
44–64	4715	35.33	943	35.33	3772	35.33	
≧65	5280	39.57	1056	39.57	4224	39.57	
**Insured premium (NT$**)							0.136
<18,000	13,111	98.25	2613	97.90	10,498	98.33	
18,000–34,999	192	1.44	49	1.84	143	1.34	
≧35,000	42	0.31	7	0.26	35	0.33	
**CAD**							<0.001
Without	12,087	90.57	2508	93.97	9579	89.72	
With	1258	9.43	161	6.03	1097	10.28	
**CVA**							<0.001
Without	12,312	92.26	2583	96.78	9729	91.13	
With	1033	7.74	86	3.22	947	8.87	
**HTN**							<0.001
Without	10,738	80.46	2247	84.19	8491	79.53	
With	2607	19.54	422	15.81	2185	20.47	
**Hyperlipidemia**							0.001
Without	12,904	96.70	2609	97.75	10,295	96.43	
With	441	3.30	60	2.25	381	3.57	
**DM**							0.031
Without	11,238	84.21	2284	85.58	8954	83.87	
With	2107	15.79	385	14.42	1722	16.13	
**ESRD**							-
Without	13,345	100.00	2669	100.00	10,676	100.00	
With	0	0.00	0	0.00	0	0.00	
**Asthma**							0.001
Without	13,098	98.15	2640	98.91	10,458	97.96	
With	247	1.85	29	1.09	218	2.04	
**CCI_R**	0.47 ± 0.81		1.27 ± 0.77		0.27 ± 0.69		<0.001
**Season**							<0.001
Spring (Mar–May)	3466	25.97	628	23.53	2838	26.58	
Summer (Jun–Aug)	3264	24.46	674	25.25	2590	24.26	
Autumn (Sep–Nov)	3019	22.62	696	26.08	2323	21.76	
Winter (Dec–Feb)	3596	26.95	671	25.14	2925	27.40	
**Location**							<0.001
Northern Taiwan	4850	36.34	835	31.29	4015	37.61	
Middle Taiwan	3674	27.53	462	17.31	3212	30.09	
Southern Taiwan	3744	28.06	1,005	37.65	2739	25.66	
Eastern Taiwan	1014	7.60	366	13.71	648	6.07	
Outlets islands	63	0.47	1	0.04	62	0.58	
**Urbanization level**							<0.001
1 (The highest)	4030	30.20	762	28.55	3268	30.61	
2	5998	44.95	1328	49.76	4670	43.74	
3	1202	9.01	360	13.49	842	7.89	
4 (The lowest)	2115	15.85	219	8.21	1896	17.76	
**Level of care**							<0.001
Hospital center	4019	30.12	685	25.67	3334	31.23	
Regional hospital	5786	43.36	1374	51.48	4412	41.33	
Local hospital	3540	26.53	610	22.86	2930	27.44	

*P*: Chi-square/Fisher exact test on category variables and t-test on continue variables. HPI: *Helicobacter pylori* infection; NT$: New Taiwan dollar; CAD: coronary artery disease; CVA: cerebrovascular accident; HTN: hypertension; DM: diabetes mellitus; ESRD: end-stage renal disease; CCI_R: Charlson comorbidity index removed coronary artery disease, cerebrovascular accident, hypertension, hyperlipidemia, diabetes mellitus, end-stage renal disease, and asthma.

**Table 2 ijerph-17-03737-t002:** Factors of irritable bowel syndrome (IBS) by using Cox regression.

Variables	Crude HR	95% CI	95% CI	*P*	Adjusted HR	95% CI	95% CI	*P*
**HPI**								
Without	Reference				Reference			
With	2.887	1.801	4.626	<0.001	3.108	1.934	4.995	<0.001
**Gender**								
Male	0.873	0.539	1.412	0.579	1.075	0.474	1.268	0.310
Female	Reference				Reference			
**Age group (years)**								
<44	Reference				Reference			
44–64	0.877	0.457	1.683	0.693	1.112	0.566	2.187	0.757
≧65	0.700	0.378	1.295	0.256	0.817	0.416	1.605	0.557
**Insured premium (NT$**)								
<18,000	Reference				Reference			
18,000–34,999	0.000	-	-	0.507	0.000	-	-	0.966
≧35,000	0.000	-	-	0.777	0.000	-	-	0.987
**CAD**								
Without	Reference				Reference			
With	0.994	0.409	1.952	0.778	1.194	0.535	2.663	0.665
**CVA**								
Without	Reference				Reference			
With	2.279	1.268	4.094	0.006	3.121	1.670	5.834	<0.001
**HTN**								
Without	Reference				Reference			
With	1.537	0.298	2.094	0.285	1.552	0.286	2.065	0.077
**Hyperlipidemia**								
Without	Reference				Reference			
With	1.042	0.506	2.894	0.366	1.048	0.561	3.317	0.432
**DM**								
Without	Reference				Reference			
With	1.462	0.921	2.965	0.054	1.488	0.729	2.042	0.064
**ESRD**								
Without	Reference				Reference			
With	-	-	-	-	-	-	-	-
**Asthma**								
Without	Reference				Reference			
With	1.475	0.361	6.018	0.588	1.809	0.433	7.550	0.416
**CCI_R**	0.905	0.775	1.058	0.209	1.017	0.875	1.184	0.309
**Season**								
Spring	Reference				Reference			
Summer	2.068	0.668	2.942	0.829	2.142	0.626	2.886	0.665
Autumn	1.532	0.573	2.037	0.064	1.543	0.578	2.147	0.074
Winter	1.389	1.175	1.864	0.020	1.413	1.186	1.919	0.030
**Location**								
Northern Taiwan	Reference							
Middle Taiwan	2.289	1.238	4.231	0.008				
Southern Taiwan	1.532	0.788	2.980	0.209				
Eastern Taiwan	1.893	0.779	4.605	0.159				
Outlets islands	0.000	-	-	0.968				
**Urbanization level**								
1 (The highest)	2.294	1.190	4.423	0.013	1.536	0.259	1.701	0.092
2	1.378	0.535	3.551	0.507	1.331	0.291	1.696	0.084
3	1.092	0.581	2.052	0.785	0.581	0.235	1.438	0.240
4 (The lowest)	Reference				Reference			
**Level of care**								
Hospital center	2.514	1.280	4.937	0.007	1.790	0.498	2.698	0.396
Regional hospital	1.690	0.890	3.212	0.109	1.590	0.275	2.224	0.177
Local hospital	Reference				Reference			

HR: hazard ratio; CI: confidence interval; HPI: *Helicobacter pylori* infection; NT$: New Taiwan dollar; CAD: coronary artery disease; CVA: cerebrovascular accident; HTN: hypertension; DM: diabetes mellitus; ESRD: end-stage renal disease; CCI_R: Charlson comorbidity index removed coronary artery disease, cerebrovascular accident, hypertension, hyperlipidemia, diabetes mellitus, end-stage renal disease, and asthma. Adjusted HR: Adjusted variables are listed. *P*: Chi-square/Fisher exact test on category variables and t-test on continue variables.

**Table 3 ijerph-17-03737-t003:** Factors of irritable bowel syndrome (IBS) stratified by variables listed in the table by using Cox regression.

	With HPI				Without HPI			With vs. Without			
	Event	PYs	Rate	Event	PYs	Rate	Ratio	Adjusted HR	95% CI	95% CI	*P*
**Total**	31	28,376.11	109.25	39	115,278.73	33.83	3.229	3.108	1.934	4.995	<0.001
**Gender**											
Male	19	18,081.31	105.08	24	74,553.61	32.19	3.264	3.228	1.487	7.010	0.003
Female	12	10,294.81	116.56	15	40,725.13	36.83	3.165	3.074	1.678	5.629	<0.001
**Age group (years)**											
<44	7	5265.65	132.94	8	20,134.93	39.73	3.346	3.269	1.130	9.458	0.029
44–64	13	8407.04	154.63	10	34,593.43	28.91	5.349	4.971	2.148	11.503	<0.001
≧65	11	14,703.42	74.81	21	60,550.38	34.68	2.157	2.201	1.506	4.589	0.035
**Insured premium (NT$**)											
<18,000	31	27,747.83	111.72	39	113,542.93	34.35	3.253	3.108	1.934	4.995	<0.001
18,000–34,999	0	624.29	0.00	0	1380.70	0.00	-	-	-	-	-
≧35,000	0	4.00	0.00	0	355.11	0.00	-	-	-	-	-
**CAD**											
Without	26	25,383.09	102.43	34	102,503.61	33.17	3.088	1.654	0.246	11.141	0.605
With	5	2993.02	167.06	5	12,775.13	39.14	4.268	3.268	1.986	5.378	<0.001
**CVA**											
Without	22	25,674.01	85.69	29	103,869.28	27.92	3.069	2.475	1.427	5.098	0.039
With	9	2702.10	333.07	10	11,409.45	87.65	3.800	3.605	2.127	6.110	<0.001
**HTN**											
Without	24	20,486.19	117.15	34	83,677.62	40.63	2.883	2.775	1.639	4.697	<0.001
With	7	7889.93	88.72	5	31,601.12	15.82	5.607	4.702	1.463	15.116	<0.001
**Hyperlipidemia**											
Without	31	27,470.42	112.85	38	111,100.13	34.20	3.299	3.186	1.977	5.133	<0.001
With	0	905.70	0.00	1	4178.61	23.93	0.000	0.000	-	-	0.752
**DM**											
Without	25	22,236.72	112.43	37	90,656.96	40.81	2.755	2.655	1.592	4.427	<0.001
With	6	6139.39	97.73	2	24,621.78	8.12	12.031	15.952	2.797	90.961	<0.001
**ESRD**											
Without	31	28,376.11	109.25	39	115,278.73	33.83	3.229	3.108	1.934	4.995	<0.001
With	0	0.00	-	0	0.00	-	-	-	-	-	-
**Asthma**											
Without	30	27,885.94	107.58	38	113,027.76	33.62	3.200	3.050	1.884	4.937	<0.001
With	1	490.17	204.01	1	2250.98	44.43	4.592	4.497	0.009	12.589	0.747
**Season**											
Spring	7	5786.86	120.96	12	24,237.41	49.51	2.443	2.197	0.847	5.700	0.105
Summer	12	6943.36	172.83	13	28,874.00	45.02	3.839	4.035	1.809	8.999	0.001
Autumn	8	9022.10	88.67	9	34,938.86	25.76	3.442	3.594	1.367	9.447	0.009
Winter	4	6623.80	60.39	5	27,228.47	18.36	3.289	2.658	0.685	10.312	0.158
**Urbanization level**											
1 (The highest)	9	7304.40	123.21	6	32,885.51	18.25	6.753	6.726	2.272	19.907	0.001
2	13	14,376.62	90.42	14	51,741.11	27.06	3.342	3.069	1.434	6.568	0.004
3	2	2293.89	87.19	4	9283.94	43.09	2.024	1.936	0.336	11.480	0.453
4 (The lowest)	7	4401.19	159.05	15	21,368.18	70.20	2.266	1.947	0.779	4.865	0.154
**Level of care**											
Hospital center	7	8402.69	83.31	6	36,328.46	16.52	5.044	5.402	1.774	16.446	0.003
Regional hospital	17	13,757.36	123.57	16	52,465.12	30.50	4.052	3..462	1.735	6.909	<0.001
Local hospital	7	6216.06	112.61	17	26,485.15	64.19	1.754	1.590	0.645	3.917	0.313

PYs: Person-years; Adjusted HR: Adjusted Hazard ratio; CI: confidence interval; HPI: *Helicobacter pylori* infection; CAD: coronary artery disease; CVA: cerebrovascular accident; HTN: hypertension; DM: diabetes mellitus; ESRD: end-stage renal disease. Adjusted for the variables listed in Table 2.

**Table 4 ijerph-17-03737-t004:** Factors of irritable bowel syndrome (IBS) among study population and PHI cohort by using Cox regression.

				HPI Cohort and Comparison Cohort				HPI Cohort			
Study Population	Event	PYs	Rate	Adjusted HR	95% CI	95% CI	*P*	Adjusted HR	95% CI	95% CI	*P*
Without HPI	39	28,376.11	137.44	Reference							
With HPI	31	28,376.11	109.25								
without medication	25	16,605.68	150.55	4.160	2.508	6.900	<0.001	Reference			
with medication	6	11,770.44	50.98	0.916	0.241	2.590	0.344	0.464	0.148	0.963	0.037

PYs: Person-years; Adjusted HR: Adjusted Hazard ratio; HPI: *Helicobacter pylori* infection. Adjusted for the variables listed in Table 2; CI: confidence interval.

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
