# Peer review of "Risk for Irritable Bowel Syndrome in Patients with Helicobacter Pylori Infection: A Nationwide Population-Based Study Cohort Study in Taiwan"

_ijerph, 2020, doi:10.3390/ijerph17103737_

Round 1

Reviewer 1 Report

The paper deals with a potentially interesting issue. The topic is not entirely new because  some papers facing with  this  issue are already available  

The strengths of this paper are the large sample of patients enrolled and the appropriate statistical analysis.

-In the introduction, when the authors state that “The pathophysiology of IBS is multifactorial and remains uncertain”, they could briefly discuss the information coming from  the paper “Irritable bowel syndrome: a disease still searching for pathogenesis, diagnosis and therapy” WJG 2014.

In the Introduction the authors state that “A Danish nationwide cohort study demonstrated that 53 there was more than an eightfold increased risk of colon cancer and a fivefold increased risk of rectal 54 cancer in the first 3 months after an IBS diagnosis”  They should precise that IBS is not a risk factor for colon cancer!

-It is not clear how the authors diagnosed IBS. Was IBS a diagnosis directly obtained from database?  The authors should explain it

- IBS symptoms and their severity are not at all described .

-A distinction of IBS subtypes  (IBS-D, IBS-C, IBS-M) would have been very interesting: there were some differences?. One could think that IBS-D and IBS-C could have important differences in their symptoms after an antibiotic therapy. The authors  correctly state that this  is  a limitation of their study  but it could be interesting if they briefly discuss a  hypothesis about possible differences of behavior of different type of IBS.

-Does CVA stand for cerebrovascular accident? It should be spelled out in the text. 

-and CAD?  There are some abbreviations needing to be clarified

-Why patients with CVA had a decreased risk of developing IBS than those without  HTN? Did the authors have an opinion?

- “In the season group analysis, patients infected  with H. pylori in the summer and autumn groups showed an increased risk of developing IBS compared with those infected in the winter and spring” How did the authors succeeded in establishing the seasonality?

-“This can also strengthen the explanation that H. pylori infection is one of the aetiologies of IBS” It seems a somewhat risky hypothesis.

-Also in the conclusions: “It seems that H. pylori infection is one of the pathogenic factors of IBS” it’ s a really risky statement!

ting if they briefly discuss a  hypothesis about possible differences of behavior of different type of IBS.

-Does CVA stand for cerebrovascular accident? It should be spelled out in the text. 

-and CAD?  There are some abbreviations needing to be clarified

-Why patients with CVA had a decreased risk of developing IBS than those without  HTN? Did the authors have an opinion?

- “In the season group analysis, patients infected  with H. pylori in the summer and autumn groups showed an increased risk of developing IBS compared with those infected in the winter and spring” How did the authors succeeded in establishing the seasonality?

-“This can also strengthen the explanation that H. pylori infection is one of the aetiologies of IBS” It seems a somewhat risky hypothesis.

-Also in the conclusions: “It seems that H. pylori infection is one of the pathogenic factors of IBS” it’ s a really risky statement!

Reviewer 2 Report

I read with interest the study titled: "Risk for Irritable Bowel Syndrome
in Patients with 3 Helicobacter Pylori Infection: The Changes in Treatment", which has several strenghts including large sample size drawn from nationwide database with a coverage rate of almost all patients’ follow-up in Taiwan. However also important limitations, not only retrospective character of the study but also potential biased from unknown confounders and errors of primary records. It is not known on which base the IBS diagnosis was made - was it Rome II criteria all any other criteria. It is likely that under IBS diagnosis other functional syndrome was included (dyspepsia ?). As NHIRD does not report information such as clinical symptoms, signs, or laboratory data, which may help distinguish different types of IBS. Also  other factors that could cause IBS, such as malignancy, inflammatory bowel disease, neurogenic bowel disease or pancreatitis, were not taken into consideration - these entities could have been responsible for alarm symptoms excluding the diagnosis of IBS. Overall number of IBS patients was low.

However this study based on it large representative sample size in Taiwan could be of interest to readers and would recommend it for publication. Nevertheless the title needs amendmend - ""Risk for Irritable Bowel Syndrome in Patients with 3 Helicobacter Pylori Infection: The Changes in Treatment"

What do authors mean by ... The changes in Treatment ?

Some minor correction: 

Table 2. Factors of Irritable Sowel syndrome by using Cox regression
Irritable Sowel should be Irritable bowel

Author Response

Response to Reviewer 2 Comments

Point 1: However also important limitations, not only retrospective character of the study but also potential biased from unknown confounders and errors of primary records. It is not known on which base the IBS diagnosis was made - was it Rome II criteria all any other criteria. It is likely that under IBS diagnosis other functional syndrome was included (dyspepsia ?). As NHIRD does not report information such as clinical symptoms, signs, or laboratory data, which may help distinguish different types of IBS.

Response 1: Thank you for your comments. In Taiwan, the diagnosis of IBS (ICD-9-CM code: 564. 1) were made by board-certified gastroenterologists, which was diagnosed by the serial versions of Rome criteria [23]. The NHI Administration randomly reviews the records of 1 in 100 ambulatory care visits and 1 in 20 inpatient claims to verify the accuracy of the diagnosis [24]. Endeavors to validate diagnosed codes or to develop methodologic approaches to address unmeasured confounders have largely increased the reliability of NHIRD studies [22]. Not all data were recorded in the NHIRD, and we were unable to evaluate the severity, clinical symptom, laboratory parameters, or stool type in IBS patients.

We have modified the statement as above into our MATERIALS AND METHODS. Please see the revised manuscript on page 3. And we had mentioned this limitation in our DISCUSSION.

Point 2: Also other factors that could cause IBS, such as malignancy, inflammatory bowel disease, neurogenic bowel disease or pancreatitis, were not taken into consideration - these entities could have been responsible for alarm symptoms excluding the diagnosis of IBS. Overall number of IBS patients was low.

Response 2: Thanks for your suggestion. Due to the many different diseases and factors of high risk of developing IBS, such as malignancy, inflammatory bowel disease, neurogenic bowel disease or pancreatitis, there are relatively small case numbers in the IBS group. This is the point that we wanted to focus on the pure population infected with H. pylori of developing IBS and increase the purity and uniformity in this study. We tried to clarify this impact.

This one million dataset is longitudinal randomized selected from National Health Insurance Research Database (NHIRD) which contained about 23 million data. Therefore, despite the dataset number is small, the actual sample can be multiplied by 23 times.

Point 3: However, this study based on it large representative sample size in Taiwan could be of interest to readers and would recommend it for publication. Nevertheless the title needs amendmend - ""Risk for Irritable Bowel Syndrome in Patients with 3 Helicobacter Pylori Infection: The Changes in Treatment"

What do authors mean by ... The changes in Treatment ?

Response 3: Thank you for your comments. We have modified the title as “Risk for Irritable Bowel Syndrome in Patients with Helicobacter Pylori Infection: A nationwide population-based study cohort study in Taiwan” into our article. Please see the revised manuscript on page 1.

Point 4: Some minor correction:

Table 2. Factors of Irritable Sowel syndrome by using Cox regression

Irritable Sowel should be Irritable bowel

Response 2: Thank you for your comments. We have modified these words into our article. Please see the revised manuscript on page 6,8.

Round 2

Reviewer 1 Report

The authors positively answered all the  questions/comments I had raised